# Partial Characterization of Three Bacteriophages Isolated from Aquaculture Hatchery Water and Their Potential in the Biocontrol of *Vibrio* spp.

**DOI:** 10.3390/microorganisms12050895

**Published:** 2024-04-29

**Authors:** İhsan Yaşa, Serap Evran, Asiye Esra Eren Eroğlu, Cengiz Önder, Maryam Allahyari, Gülçin Menderes, Müberra Kullay

**Affiliations:** 1Basic and Industrial Microbiology Section, Biology Department, Faculty of Science, Ege University, 35100 Izmir, Türkiye; asiye.esra.eren@ege.edu.tr; 2Department of Biochemistry, Faculty of Science, Ege University, 35100 Izmir, Türkiye; serap.evran@ege.edu.tr; 3Kılıç Seafood Juvenile Fish Adaptation and Hatchery Facility, 09320 Aydın, Türkiye; cengizonder@kilicdeniz.com.tr (C.Ö.); gulcinturan@kilicdeniz.com.tr (G.M.); muberrakullay@kilicdeniz.com.tr (M.K.); 4Department of Biotechnology, Graduate School of Natural and Applied Sciences, Ege University, 35100 Izmir, Türkiye; maryam.allahayry@gmail.com

**Keywords:** anti-biofilm, aquaculture, LNA-probe, phage therapy, *Vibrio* spp.

## Abstract

Bacteriophages are currently considered one of the most promising alternatives to antibiotics under the ‘One Health’ approach due to their ability to effectively combat bacterial infections. This study aimed to characterize *Vibrio* species in hatchery water samples collected from an aquaculture farm and investigate the biocontrol potential of their bacteriophages. *Vibrio* spp. (*n* = 32) isolates confirmed by LNA probe-based qPCR were used as hosts. Three *Vibrio* phages were isolated. IKEM_vK exhibited a broad host range, infecting *V. harveyi* (*n* = 8), *V. alginolyticus* (*n* = 2), *V. azureus* (*n* = 1), and *V. ordalii* (*n* = 1). IKEM_v5 showed lytic activity against *V. anguillarum* (*n* = 4) and *V. ordalii* (*n* = 1), while IKEM_v14 was specific to *V. scophtalmi* (*n* = 4). The morphological appearance of phages and their lytic effects on the host were visualized using scanning electron microscopy (SEM). All three phages remained relatively stable within the pH range of 6–11 and up to 60 °C. The lytic activities and biofilm inhibition capabilities of these phages against planktonic *Vibrio* cells support their potential applications in controlling vibriosis in aquaculture systems.

## 1. Introduction

The aquaculture sector is one of the fastest-growing industries globally. It shows promise in meeting the demand for animal protein-based foods, given the increasing human population [1]. However, bacterial infectious diseases have become a major challenge for aquaculture due to intensive production practices [2,3]. Farms situated in coastal areas, where the average annual temperature ranges between 20 °C and 26 °C, offer significant advantages for aquaculture. Unfortunately, these temperature ranges also facilitate the outbreaks of vibriosis caused by *Vibrio* spp. [4].

*Vibrio* spp. are non-spore-forming, motile, curved, Gram-negative, halophilic, facultative anaerobic bacteria [5]. They inhabit oceans, estuaries, and the internal organs of aquatic organisms, as well as environments where freshwater fish farming is practiced. Over ten species of *Vibrio*, including *V. harveyi*, *V. alginolyticus*, *V. parahaemolyticus*, *V. anguillarum*, *V. vulnificus*, and *V. ponticus*, have been reported as causative agents of diseases in numerous countries. Proficient swimmers, *Vibrio* spp. can adhere to other organisms living in water, thereby possessing the potential to infect various fish hosts [6]. Furthermore, they are often associated with the development of acute gastroenteritis in humans who consume raw or undercooked contaminated seafood, particularly shellfish [7,8].

Antibiotic-resistant *Vibrio* spp. pose a major concern [9]. Several studies in recent years have demonstrated that *Vibrio* spp. exhibit resistance to eight or more antibiotics commonly used for treating vibriosis [10,11,12]. Moreover, *Vibrio* spp. are known to form significant biofilms. However, antibiotics have been found to have minimal effect on controlling biofilm formation [13]. Bacteria within the biofilm can transfer resistance genes more readily than planktonic cells, thereby increasing the risk of antibiotic resistance dissemination. Consequently, it is imperative to develop and implement new alternatives targeting antibiotic-resistant bacteria and/or biofilms [14].

Bacteriophages, also known as bacterial viruses, present promising alternatives for microbial control compared to chemotherapeutic agents [15]. They offer advantages such as pathogen specificity, high efficiency, and minimal harm to natural microbiota compared to antibiotics [16]. Phages have undergone extensive research as antimicrobial agents in various fields, including medicine, veterinary science, agriculture, aquaculture, and food science. The efficacy of phage therapy relies on the phage’s ability to colonize infected tissues and propagate lytically within them. Before being included in the therapeutic application, each newly isolated phage must undergo rigorous characterization. This process includes determining the phage’s latent period, burst size, host range, multiplicity of infection (MOI), and phage and plaque morphology. If a phage is a candidate for potential therapeutic use, its entire genome should be sequenced. To prevent uncontrolled gene transfer during the therapy process, it is important to avoid lysogenic phages. Additionally, it is crucial to ensure that phage genomes do not contain any virulence factors or antibiotic-resistance genes that could be transferred to bacteria [17,18].

The feasibility of phage therapy as an ideal method for microbial control, even in sensitive environments such as hatcheries and adaptation facilities, is evidenced by the occurrence of autonomous transfers among animals following the initial application in aquaculture trials [19]. Promising results have been reported in recent years regarding the application of phage therapy as an alternative method to control pathogenic bacteria in aquaculture. A new jumbo phage, which is effective against *V. coralliilyticus* infections and holds potential for phage therapy against coral diseases, has been identified [20]. In one study, the application of *Vibrio* phages to shrimp resulted in a 70% higher survival rate compared to the disease control group [21]. A study was conducted to evaluate the efficacy of vB_Pd_PDCC-1 phage, which has a broad host range, during the early development of yellowtail amberjack (*Seriola rivoliana*), and it was reported that the phage had positive effects at the larval stage [22]. In another study, it was demonstrated that feeding shieldfish with a phage cocktail led to a significant reduction in the numbers of *Vibrio* and *Edwardsiella* species in their intestines. Furthermore, phage cocktail feeding was reported to effectively prevent bacterial dysbiosis in shieldfish [23].

The economic feasibility and sustainability of an aquaculture industry depend largely on the effectiveness of disease management strategies. In this regard, our focus was on utilizing bacteriophages that infect *Vibrio* as potential alternative control agents for pathogens causing infections and losses in aquaculture. In this study, three new bacteriophages were isolated from the hatchery water of sea bream (*Dicentrarchus labrax*). The study aimed to characterize *Vibrio* phages and assess their potential as biocontrol agents against different *Vibrio* species. The efficiency of all three phages in controlling pathogens and biofilms under in vitro conditions supports their use for in vivo applications.

## 2. Materials and Methods

### 2.1. Isolation and Identification of Vibrio *spp.*

From 2019 to 2023, a total of 45 water samples were collected from an aquaculture facility located along the coastline of Aydın province. The samples were taken from tanks where disease was detected during routine pathological examinations conducted by the fish health unit. Membrane filtration method was applied to isolate *Vibrio* species from samples. Each water sample was passed through 0.45 nm membrane filters (GVS, 1215676) [24]. The filter papers were placed on Thiosulphate-Citrate-Bile-Sucrose (TCBS, Merck) agar, a selective medium. Then, petri dishes were incubated at 27 °C for 24 to 48 h. Colonies with different morphotypes were subcultured in TCBS agar. The pure colonies were then examined by Gram staining, with particular attention paid to Gram-negative rod- and comma-shaped presumptive isolates. These isolates were preserved in a 25% glycerol solution at −20 °C for further analysis and bacteriophage isolation.

Genomic DNA extraction was performed from pure isolates grown in LB broth (containing 1.5% NaCl) using the High Pure Preparation kit (Roche Applied Science, Mannheim, Germany) according to the manufacturer’s instructions. DNA concentration was checked using Qubit™ 4 (Invitrogen, New York, NY, USA). It was ensured that the DNA concentration for Real-Time PCR (qPCR) was greater than 30 µg/mL.

The design and control of Locked Nucleic Acid (LNA) based TaqMan probe targeting *Vibrio* spp. was conducted through the utilization of the Clustal W Alignment program and Oligo 7 software. The probes were labeled FAM, serving as a fluorescent dye. To ensure the specificity of the primers and probes at the genus level, a verification process was carried out using BLAST within the GenBank database. Subsequently, the synthesized primers and probes were obtained from MicroSynth, Balgach, Switzerland.

The primer and probe sequences specific to the genera used are shown in Table 1. Real-time PCR assays were performed on a LightCycler 96 (Roche, Indianapolis, IN, USA) using the following reaction mix: 10 μL 2 × LightCycler 480 Probes Master mix (Roche, Indianapolis, IN, USA), 0.5 μL of each primer (10 μM stock solution), 0.2 μL probe, 3.8 μL ddH_2_O, and 5 μL DNA. The PCR cycling conditions were as follows: initial denaturation at 95 °C for 10 min, then 40 cycles of 95 °C for 10 s, and 60 °C for 60 s. *V. anguillarum* (ATCC 14181) strain was used as the positive control, while *P. damselae* subsp. piscicida (ATCC 51736), *E. coli* (ATCC 10231), and *P. aureginosa* (ATCC 27853) strains were used as negative controls.

The 16S rRNA gene was amplified using the 27 F and 1492 R primers [26]. The PCR products were visualized using gel electrophoresis and sequenced by Letgen Biotech Co., Ltd. (Izmir, Türkiye). The resulting sequences were compared to the 16S rRNA gene in the NCBI database using BLAST. Finally, the sequences of each isolate were uploaded to GenBank. 

### 2.2. Isolation of Vibrio Infecting Phages

Water samples were collected periodically from the aquaculture waters where *Vibrio* spp. were isolated. The samples were placed in 500 mL sterile bottles and transported to the laboratory while maintaining the cold chain. Fifteen *Vibrio* isolates, one from each species listed in Appendix A and confirmed by qPCR, were used as target bacteria for phage isolation. To augment hypothetical phages in water samples, 1 mL of the target bacterial culture and 5 mL of water samples were initially incubated for 48 h in conical flasks containing 50 mL of 2× Tryptic Soy Broth (TSB) with 1.5% NaCl. Following incubation, the enrichment media were centrifuged for 20 min at 8000 rpm, and the resulting supernatants were passed through 0.22 nm syringe tip cellulose acetate (CA) filters (GVS, 1213641) and subsequently transferred to new sterile falcons.

The Double-Layer Agar (DLA) method was employed to analyze phage plaques [27]. Filtrates were diluted in phage buffer (200 mM NaCl, 10 mM MgSO_4_, 50 mM Tris-HCl, pH 7.5) across a range of 10^1^–10^8^. Subsequently, 200 μL of each dilution was combined with 100 μL of host bacteria in a microcentrifuge tube. The resulting phage–host suspension was then transferred to pre-prepared semi-solid agar (0.5%) cooled to 45–47 °C. This suspension was poured onto Tryptic Soy Agar (TSA) containing 1.5% agar and incubated for 24 h. Phage plaques were harvested using a sterile Pasteur pipette and resuspended in microcentrifuge tubes containing 50 μL of phage buffer. This process was iterated six times to obtain petri dishes with pure phages. The pure phages were repropagated using the previously described enrichment protocol, and their titers were determined using the DLA method. The phage lysates obtained were stored at 4 °C for further studies.

### 2.3. Determination of Phage–Host Specificity

Phage host range analysis was carried out utilizing a drop-spot plaque assay [28]. Lawn plates were prepared using overnight cultures, and 8 μL of each phage stock was individually transferred onto distinct bacterial lawns. Subsequently, the plates were incubated at 27 °C for a period of 24–48 h. The presence of plaques was visually inspected at the end of the incubation period.

### 2.4. One-Step Growth Curve

The host culture was adjusted to an optical density (OD) of 1 at 600 during the mid-exponential growth phase. Subsequently, 5 mL of phage suspension was added to 5 mL of the host culture. Samples were collected at intervals of 5 min over a period of one hour and subjected to phage titration using the DLA method following treatment with 1% (*v*/*v*) chloroform [29]. Three independent experiments were conducted. The burst size (Bs) of the phages was determined using the formula Bs = Pt/P0, where Pt represents the phage titer at the plateau phase, and P0 represents the initial infectious titer [30].

### 2.5. Effects of Temperature and pH on Phage Stability

To assess the impact of temperature and pH on phage stability, phage suspensions (1.0 × 10^8^ pfu/mL) were subjected to incubation in a water bath across a temperature range of 20 to 80 °C. Following 1 h of incubation, the phage titer was determined. Tryptic soy broth (TSB) containing 1.5% NaCl was prepared at eight distinct pH levels using 0.1 N HCl and NaOH. Phages were introduced into TSB (1.5% NaCl) in equal volumes and then incubated for 1 h at 27 °C. Subsequently, phage titers were determined after the incubation period [31].

### 2.6. Analysis of Scanning Electron Micrographs of Phage Infection

*Vibrio* spp. cells (100 µL, 10^8^ CFU/mL) were mixed with 500 µL (10^8^ PFU/mL) of the individual phage isolates and incubated at 27 °C for 6 h. The mixture was then fixed in 2.5% glutaraldehyde (diluted in sterile phosphate-buffered saline) and 1% osmium tetraoxide (Sigma Aldrich, Steinheim, Germany). Subsequently, the samples were dehydrated in a series of ethanol concentrations (30%, 50%, 70%, 80%, 90%, 96%, and 100%). Following dehydration, they were dried using a Critical Point Dryer (Leica, Istanbul, Türkiye) [32]. Finally, the samples were coated with a gold–palladium layer of 8 nm thickness before visualization with a Scanning Electron Microscope (Thermo Scientific, Istanbul, Türkiye).

### 2.7. Optimal Multiplicity of Infection (MOI) Determination

To determine the optimal multiplicity of infection (MOI), serial dilutions of host cells in exponential phase were performed and added to aliquots of a stock solution of phages. Ratios of 0.0001, 0.001, 0.01, 0.1, 1, 10, and 100 were utilized, along with 100 μL of phage and 100 μL of *Vibrio* spp. (1 × 10^8^ CFU/mL). The final mixture was then combined with 5 mL of TSB (%1.5 NaCl). The mixtures were shaken at 27 °C for 10 h. Afterward, they were centrifuged at room temperature at 8000× *g* for 5 min to remove the cells. The precipitate was discarded, and the supernatant was filtered through a 0.45 nm syringe tip CA filter. Phage titers were determined using a DLA method. The optimal MOI was determined as the one yielding the highest phage titer [33]. Three parallel experiments were conducted to validate this MOI.

### 2.8. Anti-Biofilm Activity of Phages against Vibrio *spp.*


The biofilm-forming capacities of *Vibrio* spp. in both the presence and absence of phages were assessed following the crystal violet test protocol [34]. Overnight active cultures of *Vibrio* spp. adjusted to 0.5 McFarland turbidity were inoculated into microtiter plates containing sterile 100 μL TSB (1.5% NaCl). Following inoculation, individual phages (at optimum MOI concentration) were added to wells containing target bacteria and incubated at 27 °C for 24 h. Bacterial control wells contained only sterile medium and bacterial culture, while negative control wells contained only TSB (1.5% NaCl). After the incubation period, the microplate wells were washed three times with 0.1 M phosphate-buffered saline (PBS). The biofilms were then fixed by incubation at 80 °C for 30 min. After fixation, biofilms adhering to the microplate wells were stained with 0.1% (*w*/*v*) crystal violet and incubated for 10 min. After several washes, the stained adherent biofilms were solubilized with 95% ethanol. Measurements were then taken using a microplate reader at a wavelength of 545 nm. Three separate wells were seeded for each isolate and phage, and the average of the results was used for analysis. Experiments were performed independently in triplicate.

### 2.9. Antibacterial Effects of Phages against Planktonic Vibrio *spp.*

Phage lytic activity was investigated by measuring changes in optical density absorbance to examine the in vitro lysis of bacteria, following a previous study with minor modifications. Bacterial cultures in the logarithmic phase (approximately 10^4^ CFU/mL) were distributed into the wells of a 96-well microtiter plate. Phage suspensions were added to wells containing the target bacteria at a ratio of optimum MOI, with 50 µL per well. For each analysis, a bacterial control sample inoculated with the host culture, but not with phages, and a sterile growth medium inoculated only with phages were included. The microtiter plate was incubated at 27 °C in a programmable real-time multi-mode microplate reader, and changes in absorbance at OD600 were automatically recorded at 60 min intervals over a period of 24 h [35]. 

### 2.10. Statistical Analysis

The data obtained were analyzed using the GraphPad Prism 8 (San Diego, CA, USA). The results were presented as standard deviations. Significance was determined at the 5% level (*p* < 0.05). 

## 3. Results

### 3.1. Rapid Detection of Vibrio *spp.*


A total of 68 isolates were obtained from 45 water samples analyzed using TCBS agar. Following microscopic and phenotypic observations, 26 of these samples were discarded. Rapid detection of 32 *Vibrio* spp. was achieved from the remaining 42 suspicious isolates using the LNA probe-based real-time PCR designed in this study. Additionally, fragments of the bacterial 16S rRNA gene were amplified and sequenced as an internal control for all possible isolates. The analysis of the 16S rRNA sequences of 32 *Vibrio* spp. isolates revealed the presence of the following species: *V. harveyi*, *V. anguillarum*, *V. scophthalmi*, *V. alginolyticus*, *V. fluvialis*, *V. chagasii*, *V. sinaloensis*, *V. diabolicus*, *V. mediterranei*, *V. crassostreae*, *V. azureus*, *V. qinghaiensis*, *V. ordalii*, *V. neonatus*, and *V. vulnificus* (Appendix A).

### 3.2. Isolation and Partial Characterization of Vibrio-Infecting Phages

The DLA method was used for phage isolation. Fifteen *Vibrio* isolates, confirmed by qPCR, were used as host bacteria for bacteriophage isolation. This resulted in the purification of three phages named IKEM_vK, IKEM_v5, and IKEM_v14 (Figure 1). IKEM_v5 formed clear plaques that were 2–4 mm in size, while IKEM_vK and IKEM_v14 formed smaller, turbid plaques that were 1–3 mm in size. IKEM_vK demonstrated a wider host range by infecting 12 out of 32 *Vibrio* species, while IKEM_v5 and IKEM_v14 showed plaque formation abilities in 5 and 4 isolates, respectively (Table 2).

To describe the life cycles and adsorption capabilities of all three phages, an initial phage infection process was determined by a single-step growth curve experiment. We calculated the latent period and burst size of the phages by analyzing the dynamics of free and total phage numbers. The burst time of the IKEM_vK, IKEM_v5, and IKEM_v14 phages was found to be 25, 20, and 30 min, respectively (Figure 2). All three phages remained stable at temperatures up to 40 °C and were able to withstand temperatures up to 60 °C. They also maintained their activity within the pH range of 5–10. (Figure 2D,E). At MOI 0.01, IKEM_vK achieved the highest level with 1.82 × 10^9^ PFU/mL. Meanwhile, IKEM_14 and IKEM_v5 showed the highest level at MOI 0.001 with 1.68 ×10^10^ PFU/mL and 1.46 × 10^9^ PFU/mL, respectively (Figure 3).

Scanning electron microscopy (SEM) was utilized to analyze phages and phage-exposed *Vibrio* spp. cells (Figure 4). Three phages showed different morphologies. IKEM_vK had an icosahedral head of ~57.6 nm diameter and a tail of ~108.9 nm length (Figure 4A). IKEM_v14 had a head diameter of ~46.51 nm and a tail length of ~126.9 nm (Figure 4B). IKEM_v5 showed a head diameter of ~56.9 nm and a tail length of ~105.6 nm (Figure 4C). The images also revealed bacterial lysis resulting from bacteriophage infection and the release of phage particles (Figure 4D,E). Particularly, it was observed that phage particles tended to adsorb more onto the cell surface during bacterial cell division (Figure 4F). SEM analysis confirmed the attachment of all three phage particles to their hosts and demonstrated their lytic capabilities.

### 3.3. In Vitro Phage Treatment Assays

The biofilm stability in the presence and absence of phages is detailed in Figure 5. The OD590 values of cultures treated with phages showed a significant decrease at 24 h compared to untreated control cultures.

IKEM_vK phage, possessing a wide host range, was applied to 12 isolates, including the *V. harveyi*, *V. alginolyticus*, *V. azureus*, and *V. ordalii* strains, resulting in the inhibition of biofilm formation by more than 50% in 7 of them. The highest level of biofilm inhibition, 75.9%, was observed in *V. harveyi* strain Gdp17 (Figure 5A,B). *V. scophthalmi* strains Gdp33, Gdp35, Gdp38, and Gdp39, treated with IKEM_v14, exhibited reductions in biofilm formation by 61.5%, 45.8%, 60.5%, and 39.8%, respectively (Figure 5C,D). When the growth medium of *V. anguillarum* strains Gdp19, Gdp20, Gdp30, Gdp34, and *V. ordalii* strain Gdp41 was supplemented with IKEM_v5 phage, it was able to inhibit bacterial biofilm formation to varying degrees, ranging from 33.8% to 59.9% (Figure 5E,F).

Bacterial inactivation was measured against *Vibrio* spp. hosts, where each individual phage exhibited the highest biofilm inhibition. In control samples lacking phages, it was noted that *Vibrio* spp. (*V. harveyi* strain Gdp17, *V. scophthalmi* strain Gdp33, and *V. anguillarum* strain Gdp34) proliferated during the incubation period, resulting in a concurrent rise in absorbance values (Figure 6). The phage control remained constant at all time points (ANOVA, *p* > 0.05).

Two hours after the initial inoculation, the optical densities of the Gdp17 isolate cultured alone and cultured with IKEM_vK were measured as 0.35 ± 0.03 and 0.34 ± 0.02, respectively. Although there was no significant difference in these values (ANOVA, *p* > 0.05), the optical density of the control environment containing only Gdp17 was 0.64 ± 0.02 at the 8th hour of the experiment. In contrast, the optical density of Gdp17 cultured with IKEM_vK was 0.46 ± 0.01, and this difference was statistically significant (ANOVA, *p* < 0.05). However, bacterial inactivation remained considerably high after 24 h of incubation. The optical density of Gdp17 cultured alone was measured as 1.66± 0.02, while that cultured with IKEM_vK was measured as 0.84 ± 0.01. (Figure 6).

After phage treatment with IKEM_v14, statistically significant initial results were obtained at the 12th hour of the experiment (ANOVA, *p* < 0.05). The optical densities of Gdp33 cultured alone and cultured with IKEM_v14 were recorded as 0.80 ± 0.01 and 0.51 ± 0.02, respectively. At the 24th hour of the experiment, the optical density of host cells cultured alone was 1.74 ± 0.02. In contrast, the optical density was measured as 0.65 ± 0.03 in the medium exposed to IKEM_v14. These values were found to be significantly different (ANOVA, *p* < 0.05), indicating an effect of IKEM_v14 on the host cells. (Figure 6).

Following treatment with the IKEM_v5 phage, statistically significant initial results were obtained at the 4th hour of the experiment (ANOVA, *p* < 0.05). The optical densities of Gdp34 cultured alone and co-cultured with IKEM_v5 were recorded as 0.49 ± 0.01 and 0.31 ± 0.03, respectively. By the 24th hour of the experiment, the optical density of host cells cultured alone reached 1.58 ± 0.02. Conversely, in the medium exposed to IKEM_v5, the optical density was measured to be 0.52 ± 0.01. These values showed a significant difference (ANOVA, *p* < 0.05), indicating the effect of IKEM_v5 on the host cells (Figure 6).

## 4. Discussion

Bacterial outbreaks are becoming more common in aquaculture environments, which is worsened by global warming [36]. The increasing resistance of bacteria in biofilms to antibiotics is a significant obstacle to current treatment methods that rely on antibiotics [37,38,39]. Vibriosis outbreaks caused by *Vibrio* species are devastating bacterial diseases in aquaculture due to their broad host range and high mortality rates. Due to the growing awareness of antibiotic resistance and restrictions on antibiotic use, phage applications are increasingly popular as alternatives to antibiotic treatment in animal production sectors [40,41,42]. The use of phages is particularly promising for controlling infections in aquatic environments such as aquacultures [43,44]. The effectiveness of disease management protocols is crucial for the viability of the aquaculture industry. In this study, we collaborated with industry experts to explore the potential of vibrio-targeted bacteriophages as an alternative method to control pathogens that cause harmful infections and economic losses in aquaculture systems [45].

We used LNA probe-based qPCR in the study, which greatly facilitated the identification of 32 *Vibrio* spp. Furthermore, the 16S rRNA gene of each isolate was sequenced, and an internal control was performed for all possible isolates. The real-time PCR method enabled the discrimination of members of the genus *Vibrio* in less than 4 h. LNA probes offer significant advantages such as high sensitivity, a wide linear range, a clinically relevant detection limit, ease of design, and improved signal generation [46,47]. LNA probes have the ability to distinguish single-base mutations through amplification curves without the need for complex detection or additional Tm analysis [48].

Analysis of 16S rRNA sequences from *Vibrio* species isolated from aquaculture waters showed that *V. harveyi* (25%), *V. anguillarum* (12.5%), and *V. scophthalmi* (12.5%) had higher rates compared to other *Vibrio* species. These species are already frequently isolated during vibriosis outbreaks [49,50]. *Vibrio harveyi* is frequently found in shrimp cultures, whereas *Vibrio anguillarum* is commonly associated with disease in sea bass and trout [51,52,53]. Studies suggest that *Vibrio scophthalmi* is an opportunistic pathogen that typically does not cause disease in healthy fish. However, exposure to environmental stressors can weaken fish immunity, making them susceptible to infection by this bacterial species, which can result in illness or death [54]. For instance, when the water temperature reaches 20 °C, Japanese eels (*Anguilla japonica*) are more susceptible to infection by *V. scophthalmi*, which can cause severe enteritis and ascites symptoms, ultimately resulting in high mortality rates. Moreover, *V. scophthalmi* is considered a secondary infectious agent that significantly increases the mortality rate of diseased fish following infections by other pathogens [55].

We used *Vibrio* spp. isolates as hosts and obtained three bacteriophages (IKEM_vK, IKEM_v14 and IKEM_v5) by repeated plaque purification. In discriminating between the phages, we first focused on phage plaque morphologies at the same titers. It was previously reported that transformations between clear and turbid plaques could occur depending on the phage titer. The size and morphology of phage plaques are also determined by the lysis time [56]. The IKEM_v5 phage has a shorter latent period compared to IKEM_vK and IKEM_v14, resulting in larger plaques due to a higher burst size. Previous studies have shown that an increase in burst size leads to the formation of larger plaques [57].

The host ranges of all three phages were determined by conducting spot tests on phage-treated bacterial cultures. IKEM_vK exhibited a broader host range, infecting 12 out of 32 *Vibrio* strains, including *V. harveyi*, *V. ordalii*, *V. azureus*, and *V. alginolyticus*. Meanwhile, IKEM_v5 and IKEM_v14 showed plaque formation abilities in five (*V. anguillarum*, *V. ordalii*) and four (*V. scophthalmi*) strains, respectively. The results indicate that the IKEM_vK, IKEM_v14, and IKEM_v5 isolates are distinct phages with different host ranges. Bacteriophage receptor specificity is directly associated with host range, as receptor proteins bind to specific structures on the outer surface of host bacteria [58]. In silico studies can now predict phage host ranges in advance [59]. The proper selection of bacteriophages is crucial for the success of phage-mediated protection and treatment. When selecting phages, it is important to consider criteria such as host range, burst size, survival in the usage environment, and latent period [60]. In our study, all three phages were found to be stable up to 40 °C and within a pH range of 5–10, indicating their ability to survive under normal seawater conditions (0–35 °C and pH 7.4–8.4).

The use of phages is a promising alternative to prevent and control biofilm formation, which is one of the most important virulence factors in pathogenic bacteria. Bacteria in biofilms are much more resistant to antibiotics than planktonic cells [14,61]. Our crystal violet staining experiments showed that most of the strains we isolated were strong biofilm producers. However, all three phages demonstrated biofilm inhibition capabilities ranging from 75.9% to 15.7% when added to the growth medium of *Vibrio* hosts. IKEM_vK was successful in reducing biofilm formation by over 50% in seven *Vibrio* spp. isolates. Meanwhile, IKEM_v14 provided biofilm inhibition ranging from 61.5% to 39.8% in the four strains it infected. IKEM_v5 exhibited the highest anti-biofilm activity at 59.9%. Previous studies have demonstrated the anti-biofilm effect of phages and their enzymes (enzibiotics) [62]. It was observed that the efficacy of phage FE11 in controlling biofilm formation by *V. parahaemolyticus* was directly correlated with a high concentration of phages [63]. Nonetheless, bacterial density may still be an uncontrolled variable in field applications. Hence, implementing the periodic administration of phages at a specific titer by propagating them in situ during treatment, rather than using the traditional dosing method, will enhance the success of phage-mediated protection and treatment.

In this study, we investigated the lytic activity of phages on planktonic *Vibrio* cells. The optical densities of *Vibrio* spp. cells cultured with phages were significantly lower than those of the control groups until the 20th hour of the experiment. However, bacterial growth resumed after 20 h, albeit slowly. It is important to note that over time, host bacteria may develop resistance to phages. The previous literature reported that bacterial phage resistance is associated with decreased virulence in the host [29]. To prevent the development of resistance, it is recommended to isolate several phages with different host ranges and apply them in mixtures against target pathogens [16,43,64]. However, a balance must be struck in the cocktail’s host range, as a more costly, broader lytic spectrum may encourage the undesired selection of resistant bacterial strains or kill off unintended bacteria, thus posing a risk of disrupting beneficial microbiota. An alternative approach is to utilize cocktails comprising phages that target distinct receptors on the same bacterial cell, which may assist in the retardation of the emergence of resistance [44,45,65]. The establishment of phage libraries isolated against pathogens obtained from local farms will facilitate the development of more effective phage cocktails tailored to the target region. Personalized cocktails, currently employed in human medicine, minimize the impact on normal microbiota by excluding “useless phages” [42,66]. This approach would be particularly beneficial in industrial treatments, especially in aquaculture, where large quantities of phages may be necessary.

## 5. Conclusions

Effective disease management strategies are crucial for success in animal production industries. Through industry–academic collaboration, we successfully isolated new *Vibrio* strains and their infecting phages. Our findings demonstrate the potential of these phages to combat vibriosis agents originating from local farms. However, further studies, such as full genomic characterization, are needed for the safe use of these phages in aquaculture and to prevent the transfer of pathogenic traits through phages. The objective of our future research is to continually update the phage library against vibriosis by enriching it with new isolates and developing novel phage cocktails tailored to regional application segments. While phage therapy still has some limitations, future applications utilizing phage components or phage design represent the driving force of our study. The renewed interest in phage research motivates scientists, and the data generated in this field will be important milestones for successful future applications.

## Figures and Tables

**Figure 1 microorganisms-12-00895-f001:**
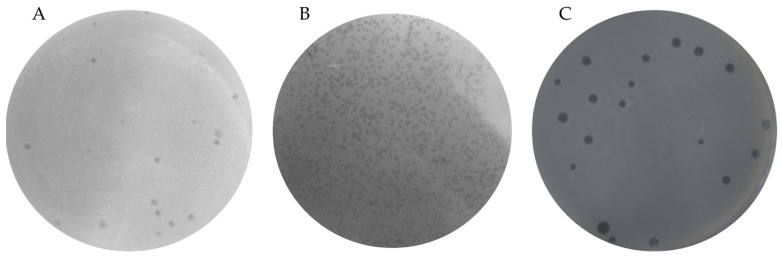
Plaques produced by phages IKEM_vK (**A**), IKEM_v14 (**B**), and IKEM_v5 (**C**).

**Figure 2 microorganisms-12-00895-f002:**
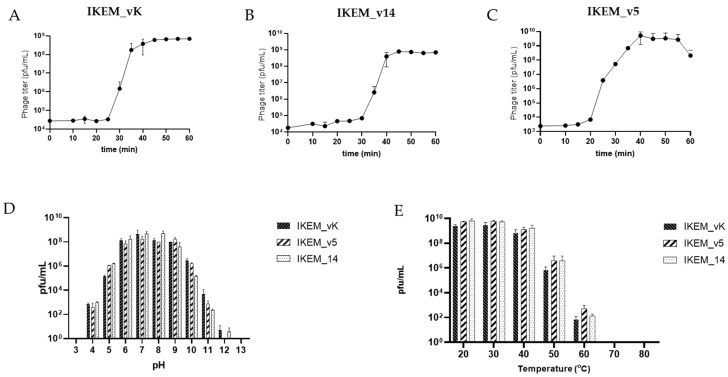
Life cycle parameters of IKEM_vK (**A**), IKEM_v14 (**B**), and IKEM_v5 (**C**) and stability of phages at several pH values (**D**) and temperatures (**E**).

**Figure 3 microorganisms-12-00895-f003:**
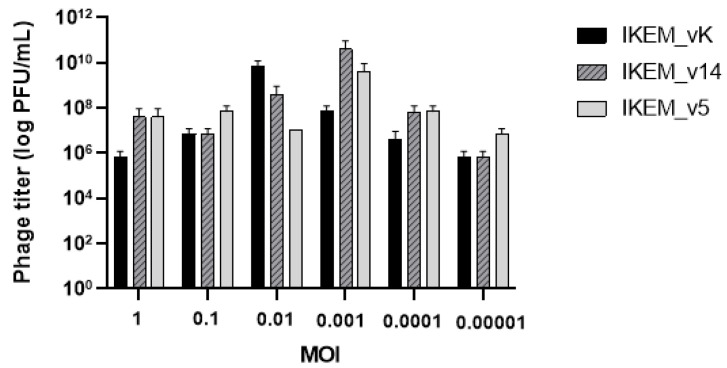
Multiplicity of infection values of IKEM_vK, IKEM_v14, and IKEM_v5 phages.

**Figure 4 microorganisms-12-00895-f004:**
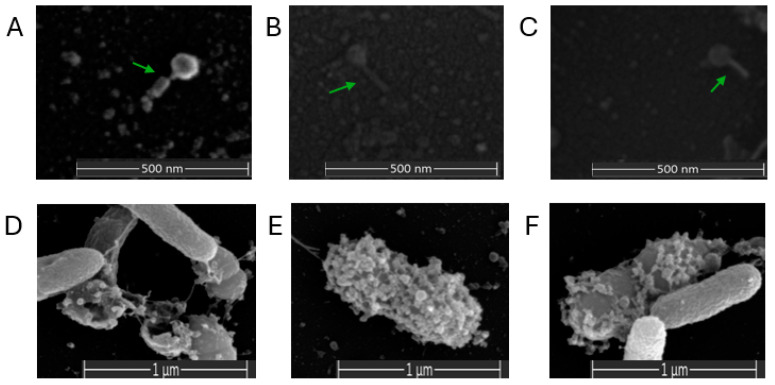
Electron microscopy images of phages IKEM_vK (**A**), IKEM_v14 (**B**), and IKEM_v5 (**C**). Image of *Vibrio harveyi* strain Gdp1 lysed by IKEM_vK (**D**). *V. anguillarum* strain Gdp30 (**E**) and *V. scophthalmi* strain Gdp33 cells surrounded by phages (**F**). Phage particles are indicated by green arrows.

**Figure 5 microorganisms-12-00895-f005:**
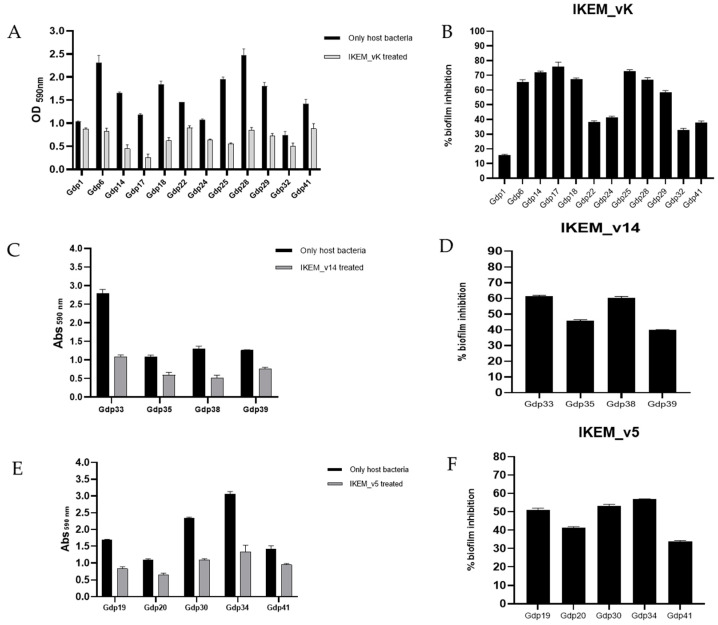
Inhibitory effect of phages on biofilm formation of *Vibrio* spp. (**A**,**C**,**E**) are the *Vibrio* biofilm production capacity measured at 590 nm wavelength in the presence and absence of IKEM_vK, IKEM_v14, and IKEM_v5, respectively. (**B**,**D**,**F**) are the percentage biofilm inhibition rates of IKEM_vK, IKEM_v14, and IKEM_v5 phage, respectively.

**Figure 6 microorganisms-12-00895-f006:**
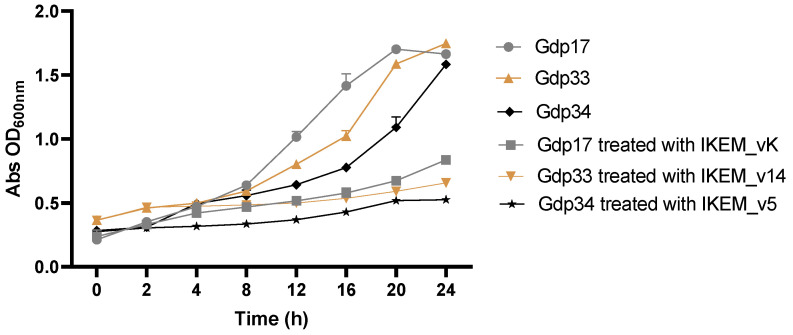
Lytic activity of phages against Gdp17, Gdp33, and Gdp34 isolates. The presented data represents the mean values of three independent experiments. Error bars represent the standard deviations.

**Table 1 microorganisms-12-00895-t001:** Primers and probe information.

Code	Sequence 5′-3′	Tm (°C)	Target	Reference
567 F	GGC GTA AAG CGC ATG CAG GT	61.4	16S rDNA	[25]
680 R	GAA ATT CTA CCC CCC TCT ACA G	54.3	16S rDNA	
Vsp P	TTA AGT CAG ATG TGA AAG CCC GGG	59.4	16S rDNA	This study

**Table 2 microorganisms-12-00895-t002:** Assessment of host range through spot testing.

Host Range	*Vibrio* Phages		
	IKEM_vK	IKEM_v14	IKEM_v5
*V. harveyi*			
Gdp1 (PP270122)	+		
Gdp6 (PP270125)	+		
Gdp14 (PP270130)	+		
Gdp17 (PP270133)	+		
Gdp22 (PP270137)	+		
Gdp25 (PP270139)	+		
Gdp28 (PP270140)	+		
Gdp29 (PP270141)	+		
*V. anguillarum*			
Gdp19 (PP270135)			+
Gdp20 (PP270136)			+
Gdp30 (PP270142)			+
Gdp34 (PP270146)			+
*V. scophtalmi*			
Gdp33 (PP270145)		+	
Gdp35 (PP270147)		+	
Gdp38 (PP270150)		+	
Gdp39 (PP270151)		+	
*V. alginolyticus*			
Gdp18 (PP270134)	+		
Gdp24 (PP270138)	+		
*V. azureus*			
Gdp32 (PP270144)	+		
*V. ordalii*			
Gdp41 (PP270152)	+		+
*V. neonatanus*			
Gdp42 (PP270153)			

## Data Availability

All sequence data supporting the findings of this study have been deposited in GenBank (https://www.ncbi.nlm.nih.gov/genbank/ (accessed on 10 February 2024)) with accession numbers PP270122-PP270153.

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
