# Peer review of "Partial Characterization of Three Bacteriophages Isolated from Aquaculture Hatchery Water and Their Potential in the Biocontrol of Vibrio spp."

_microorganisms, 2024, doi:10.3390/microorganisms12050895_

Round 1

Reviewer 1 Report

Comments and Suggestions for Authors

The reviewed manuscript of YaÅŸa et al. addresses an important problem of aquaculture biocontrol of Vibrio spp. Vibrio spp. are causative agents of diseases of aquatic organisms. Nowadays, when antibiotic resistance has become one of the major problems in the treatment of bacterial infection, the use of phages that can kill bacteria and inhibit biofilm formation has become an interesting alternative to antibiotics.The paper by Yasha et al characterizes three putatively novel and putatively lytic phages for the control of vibriosis in aquaculture systems. The design of the experiment is logical and straightforward, the data are presented convincingly, and the discussion part is comprehensive. Overall, the paper concerns an important field of phage therapy, and can be considered for publication. However, there is a serious problem associated with the lack of any genomic characterization of isolated phages. Without this characteristic it is impossible to claim that phages are new and lytic. In addition, in the introduction, authors should mention the general requirements for phages for therapeutic purposes, including their strictly lytic lifestyle and the absence of virulence factors and antibiotic resistance genes in their genomes. Authors must sequence the phages, analyze the genomes, and demonstrate the absence of genes associated with temperate lifestyles (such as integrases), and show that the phages do not belong to already known and characterized species.

Author Response

Dear reviewer,

Reviewer 2 Report

Comments and Suggestions for Authors

The manuscript by Ä°. YaÅŸa et al. describes the isolation of a set of vibrio species and specific bacteriophages of potential relevance for phage therapy in the context of aquaculture. The study is of significance due to its potential application in the prevention of multidrug-resistant vibriosis among species of economic importance in aquaculture. The manuscript is well-drafted and presented clearly and logically. Most of the references are from the last 5 years and the authors do not self-cite. The methods section provides sufficient detail for reproducibility by independent researchers. The data is well presented in the figures and interpreted appropriately throughout the manuscript. The statistical analyses were appropriate for the kind of data generated by the authors. However, the authors should address the following critiques for publication.

Major critiques:

 The conclusions of the study can be improved.  The authors may abound on the potential of the isolated phages for the treatment of diseases caused by the vibrio species isolated in the context of the local aquaculture. For example, the authors may discuss if a cocktail of the isolated phages would be appropriate to control a wide range of vibrios according to the vibrio incidence reported by the local farms; if it is desirable to isolate additional different phages, etc.

It would have been nice if the authors attempted to identify the receptors of the phages or if they at least proposed this as a future direction in the discussion section. The authors discussed appropriately the utilization of phages with different host ranges but the utilization of phages that recognize different receptors in the same bacterial species is important for the design of therapeutic phage cocktails since this helps to avoid the emergence of phage resistance. Adding some discussion about this is desirable.

Minor critiques:

The sentence in lines 338-339 (page 11) is redundant with the previous sentence in lines 337-338. Remove one of them.

Comments on the Quality of English Language

The quality of English looks adequate to me, but I am not a native speaker myself. Thus, minor editing might be required by a proficient editor.

Author Response

Dear Reviewer

Round 2

Reviewer 1 Report

Comments and Suggestions for Authors

I still think that the authors should have carried out a genomic characterization of the phages. But the study is interesting from the point of view of the results of biocontrol experiments and, I think, can be published in the journal Microorganisms.